# Genomic Analysis of Antimicrobial Resistance Genotype-to-Phenotype Agreement in *Helicobacter pylori*

**DOI:** 10.3390/microorganisms9010002

**Published:** 2020-12-22

**Authors:** Tal Domanovich-Asor, Yair Motro, Boris Khalfin, Hillary A. Craddock, Avi Peretz, Jacob Moran-Gilad

**Affiliations:** 1MAGICAL Group, Department of Health Systems Management, Faculty of Health Sciences, Ben-Gurion University of the Negev, Beer-Sheva 8410501, Israel; taldo60@gmail.com (T.D.-A.); motroy@post.bgu.ac.il (Y.M.); boriskh83@gmail.com (B.K.); hcraddock5@gmail.com (H.A.C.); 2Clinical Microbiology Laboratory, Baruch Padeh Medical Center, Poriyah and Azrieli Faculty of Medicine, Bar-Ilan University, Safed 1311502, Israel; aperetz@poria.health.gov.il

**Keywords:** whole genome sequencing (WGS), *H. pylori*, clinical antimicrobial susceptibility testing

## Abstract

Antimicrobial resistance (AMR) in *Helicobacter pylori* is increasing and can result in treatment failure and inappropriate antibiotic usage. This study used whole genome sequencing (WGS) to comprehensively analyze the *H. pylori* resistome and phylogeny in order to characterize Israeli *H. pylori*. Israeli *H. pylori* isolates (*n* = 48) underwent antimicrobial susceptibility testing (AST) against five antimicrobials and WGS analysis. Literature review identified 111 mutations reported to correlate with phenotypic resistance to these antimicrobials. Analysis was conducted via our *in-house* bioinformatics pipeline targeting point mutations in the relevant genes (*pbp1A*, 23S *rRNA*, *gyrA, rdxA, frxA*, and *rpoB*) in order to assess genotype-to-phenotype correlation. Resistance rates of study isolates were as follows: clarithromycin 54%, metronidazole 31%, amoxicillin 10%, rifampicin 4%, and levofloxacin 2%. Genotype-to-phenotype correlation was inconsistent; for every analyzed gene at least one phenotypically susceptible isolate was found to have a mutation previously associated with resistance. This was also observed regarding mutations commonly used in commercial kits to diagnose AMR in *H. pylori* cases. Furthermore, 11 novel point mutations associated with a resistant phenotype were detected. Analysis of a unique set of *H. pylori* isolates demonstrates that inferring resistance phenotypes from WGS in *H. pylori* remains challenging and should be optimized further.

## 1. Introduction

Antimicrobial resistance (AMR) is increasingly being reported among clinical *Helicobacter pylori* isolates worldwide [1,2]. In various parts of the world, disparate *H. pylori* antimicrobial resistance rates have been observed, suggesting geographical variation [3,4]. In Israel alone, six papers dealing with the prevalence of antimicrobial resistance among *H. pylori* isolates from the last decade have observed differing resistance rates [2,5,6,7,8,9], suggesting that within-country rates may vary among ethnic and social populations. Given these findings as well as the overall dearth of *H. pylori* surveillance, geographic characterization of antibiotic resistance is extremely important in order to adjust treatment appropriately [10].

However, antimicrobial surveillance data for *H. pylori* are limited as the use of antimicrobial susceptibility testing (AST) is frequently only implemented after initial treatment failure [10]. To address these challenges of variable resistance rates, as well as other logistical challenges involved in AST of *H. pylori* isolates, culture-independent techniques have been implemented. Several PCR-based commercial kits are commonly used to identify genotypic resistance from clinical samples [11], usually by targeting three to seven point mutations in the 23S *rRNA* (A2142C, A2142G, and A2143G) and *gyrA* (N87K, D91G, D91N, D91Y) genes that are generally considered to correlate with resistance [11,12]. However, these tests are limited due to imperfect correlations between these mutations and resistance [13,14,15,16], the numerous other mutations associated with resistance [17], and the possibility of co-infection with other organisms like Epstein-Barr Virus [18].

With the advent of next-generation sequencing (NGS) technology, utilizing genomics to investigate AMR in bacteria of public health importance is a growing area of research and combining whole genome sequencing (WGS) with traditional phenotypic resistance data can produce powerful results. Given the dearth of *H. pylori* AMR surveillance overall, the challenges involved in traditional *H. pylori* AST, as well as the potential for WGS to provide needed data, more research is needed in this regard. In order to address this need, we used WGS in order to conduct a comprehensive genotypic-phenotypic comparison among a set of Israeli *H. pylori* isolates.

## 2. Materials and Methods

### 2.1. H. pylori Culture and Phenotypic Antimicrobial Susceptibility Testing (AST)

Forty-eight unique patient *H. pylori* isolates were recovered from routine, clinical samples in the Baruch Padeh Medical Center Poriya in Northern Israel between 2015–2019. All biopsy specimens were placed in sterile Eppendorf tubes containing 1 mL of sterile physiological solution (0.9% NaCl) and were sent to the Clinical Microbiology Laboratory in Poriya Medical Center. Biopsies were minced manually with a sterile scalpel and seeded on Modified BD *Helicobacter* Agar (BD Diagnostics, Sparks, MD, USA) and blood agar (Hy-lab, Rehovot, Israel). The plates were incubated at 37 °C in a micro-aerophilic atmosphere produced by a gas generating system adapted for *Campylobacter* (Campy-Gen™, Oxoid, Basingstoke, UK) and inspected for growth at 7 and 10 days. *H. pylori* was identified based on Gram-staining followed by oxidase, catalase, and urease tests. Final identification was performed using MALDI-TOF MS (Bruker Daltonics, Bremen, Germany). For AST and DNA extraction one colony of the bacterial isolate was picked and re-grown on a selective agar under the conditions mentioned above. *H. pylori* suspensions were adjusted to a McFarland standard of 3.0 and subject to AST as part of a routine, clinical diagnostics procedure for five antimicrobials (amoxicillin, clarithromycin, levofloxacin, metronidazole, and rifampicin) using the gradient diffusion E-test method (bioMerieux, Mercy l’Etoile, France) and minimum inhibitory concentration (MIC) Test Strip (MTS™) (Liofilchem, Roseto degli Abruzzi, Italy). Testing was performed according to manufacturers’ instructions using Mueller-Hinton agar plates with 5% sheep’s [19,20]. Following 72h of incubation, minimum inhibitory concentration (MIC) values were determined and interpreted according to British Society for Antimicrobial Chemotherapy (BSAC) breakpoints [21]. The clinical laboratory is ISO-accredited and maintains a fully operational quality assurance and quality control scheme. Borderline MIC values were repeated and confirmed. *H. pylori* ATCC 43,504 was used for quality control. This study was approved by the Ethics Committee of the Baruch Padeh Medical Center Poriya (#23-13-POR).

### 2.2. DNA Extraction, Library Preparation, and WGS of H. pylori Isolates

DNA was extracted from isolates via a physical lysis process utilizing beating with acid-washed glass beads (Precellys Evolution, Bertin Instruments, Montigny-le-Bretonneux, France). High-quality DNA was subject to library preparation using the Nextera Flex kit (Illumina, San Diego, CA, USA) according to the manufacturer’s recommendations, followed by paired-end sequencing on Illumina iSeq 100 or Miseq platforms. Raw WGS data (FASTQ files) underwent quality control (QC), filtering, trimming, and de novo assembly using shovill (spades, v3.12; using the parameters ‘--trim’ and ‘--opts “--sc”’) (https://github.com/tseemann/shovill). Assemblies were annotated using the tool prokka (v 1.14.0; using the *H. pylori* 26,695 reference strain proteins in the parameter ‘--proteins GCF_000008525.1_ASM852v1_protein.faa’). Further details are available in Appendix A. Sequence data have been deposited to BioProject PRJEB37854.

### 2.3. Antimicrobial Resistance-Associated Mutations Analysis

A literature review was conducted to devise a list of mutations at the genes *pbp1A* (amoxicillin-Resistance (R)), 23S *rRNA* (clarithromycin-R), *gyrA* (levofloxacin-R), *rdxA* and *frxA* (metronidazole-R), and *rpoB* (rifampicin-R) (Appendix A). These genes were extracted and aligned with the tool MAFFT (v7.397) [22]. The 23S *rRNA* genes were identified using rnammer (v1.2) since rnammer provides more complete 23S *rRNA* sequences. Only perfectly-aligned gene sequences were analyzed for the presence of the abovementioned mutations (number of analysed alignments for each gene is shown in Appendix A). The isolates then underwent further analysis to compare genotypic and phenotypic resistance data and describe genotype-to-phenotype correlations. Phenotypically-resistant isolates were analyzed for any novel point mutations. Subsequently, the remaining susceptible isolates were interrogated for these novel mutations. For the MIC plots and mutations clustering analysis, R version 3.4.2 [23] was used with the following packages: tidyr_0.8.3 [24], data.table_1.12.0 [25], cowplot_0.9.4 [26], ggplot2_3.1.0 [27], pheatmap_1.0.12 [28], RColorBrewer_1.1-2 [29], gplots_3.0.1 [30] and ggtree_1.15.6 [31,32]. The tools csvtk (https://github.com/shenwei356/csvtk) and seqkit (https://github.com/shenwei356/seqkit) were also used [33]. Clustering analysis was performed on the genes in which point mutations were identified, in order to identify correlation between observed mutations and MIC values.

## 3. Results

### 3.1. Phenotypic Antimicrobial Susceptibility Testing (AST) of Study Isolates

AST was performed on all 48 isolates to assess susceptibility to the five antibiotics most commonly used for *H. pylori* treatment. The distribution of MIC values for tested agents is shown in Appendix A. Full susceptibility to all tested antimicrobials was observed in 10 isolates (20.8%). Resistance was observed with the following rates: amoxicillin 10.4%, clarithromycin 54.2%, levofloxacin 2.0%, metronidazole 31.1% and rifampicin 4.2%. Only clarithromycin has an intermediately-resistant category (0.25–0.5 mg/L), and eight isolates (16.7%) were observed to be intermediate. Resistance to more than one antimicrobial drug was observed in nine isolates, 18.8% (clarithromycin-metronidazole *n* = 3; clarithromycin-amoxicillin *n* = 2; clarithromycin-rifampicin *n* = 1; clarithromycin-metronidazole-levofloxacin *n* = 1; clarithromycin-metronidazole-rifampicin *n* = 1).

### 3.2. Resistome Analysis and Genotype-to-Phenotype Correlation of Resistance

Mutations previously associated with phenotypic resistance were compared to AST results, and 51 such mutations were detected as shown in Figure 1. Of these, 25 mutations were found among both phenotypically-susceptible and -resistant isolates, while 24 mutations were found exclusively among phenotypically-susceptible isolates. The mutations A2142G at the 23s *rRNA* gene, mutations R16H, R16C, and the variant R90S in the *rdxA* gene were found exclusively in resistant isolates. The new variant T593S at the *pbp1*A gene was detected in both susceptible and resistant isolates. Seventeen isolates had the A2143G mutation at the 23S *rRNA* gene, of which seven were resistant, four were intermediately-resistant and six were susceptible. The *gyrA* mutations N87K, and D91N were found among 4.2% of isolates, in both resistant and susceptible isolates. No clear correlation was observed between resistant phenotypes and co-existing mutations after clustering analysis (Figure 2).

### 3.3. Novel Point Mutations

Novel point mutations discovered among phenotypically-resistant isolates are summarized in Appendix A. The mutations G94E at the *pbp1A* gene, C2173T and G2212A at the 23S *rRNA* gene, T239M at the *gyrA* gene, G122R at the *rdxA* gene, A70T and A138V at the *frxA* gene were found among resistant isolates only.

## 4. Discussion

Previous resistome analysis of *H. pylori* via WGS has highlighted the role of point mutations as a mechanism of antibiotic resistance [34]. Furthermore, preliminary phenotypic-genotypic analysis utilizing WGS has strengthened the knowledge base of the association between 23S *rRNA*, *gyrA*, and *rpoB* mutations and resistance to macrolides, fluoroquinolones, and rifampicin but the practical implications deserve further study [34,35,36]. For example, Lauener et al. [35] noted a strong correlation between certain mutations in the 23 *rRNA* gene (A2146C, A2146G, and A2147G) and resistance, however these mutations were not observed in our study. This further supports the need for future research on various mutations and their phenotypic correlation. Our results add to these findings by demonstrating a complicated relationship between genotype and phenotype where mutations previously associated with resistance were found among phenotypically-susceptible isolates. For example, we observed discrepancies among mutations frequently associated with resistance including the A2143G mutation [12]; 10 out of 17 (59%) isolates harboring that mutation were susceptible (*n* = 6) or intermediately-resistant (*n* = 4). While this mutation is highly utilized in PCR-based kits, the discrepancy observed in this study has been observed, at differing rates, in previous PCR-based studies [13,14,15,16,37,38]. Similarly, further inconsistency was observed among 48 other mutations. These findings suggest that further research is needed to accurately assess phenotype-genotype correlation in this microorganism. It also bears noting that recent research has found that *H. pylori* can sometimes transform from the expected spiral form to a coccoid form, which could result in an inaccurate AST if the bacteria are in a viable-but-nonculturable state. This transformation could result in treatment failure without the presence of ARGs, further complicating the picture of AMR in *H. pylori* [39]. While WGS may be useful in overcoming the limitations of studies by allowing analysis of the full resistome, the relative contribution of each mutation to the MIC as well as the effect of combinations of mutations and shape transformation remains to be elucidated.

Regarding phenotypic antimicrobial resistance, we observed that resistance rates are higher for clarithromycin (54.2%) and metronidazole (31.1%) than amoxicillin (10.4%), rifampicin (4.2%) and levofloxacin (2%). When comparing the results of this study to previous studies of resistance rates in *H. pylori* in Israel, findings varied based on antibiotic as well as ethnic groups. Amoxicillin resistance rates were consistent with the other studies [2,8,9]. Clarithromycin resistance rates were similar to the rate observed among an Arab-Israeli population in Yeganeh et al. [8] article and higher compared to the Jewish-Israeli population in the same study as well as other studies [2,5,7,8,9]. Levofloxacin resistance rates in our study were lower than those previously observed by Peretz et al. [6] but similar to those of Pastukh et al. [2]. Observed metronidazole resistance rates were similar to those of Zevit et al. [5] and are within the range of rates observed among all studies [2,7,8,9]. For rifampicin, resistance rates are closer to those of the Arab-Israeli patient population than the Jewish-Israeli patient population in the Yeganeh et al. [8] article. It is not surprising that several of our findings are similar to previous Israeli findings, as some of the previous studies were carried out in the same hospital which contributed isolates to the current study, and therefore are likely from a similar patient pool experiencing similar exposures and antibiotic pressures. The instances where our rates differ are not substantial, and this variation may be due to factors including but not limited to sample size, differential antibiotic exposure among different ethnic and social groups, as well as acquisition of different *H. pylori* strains during international travel. A large proportion of Israeli citizens and residents are immigrants from other countries, including Eastern Europe, and research has noted that prevalence of *H. pylori* is high in immigrants from high-prevalence countries [40]. Patient ethnicity was not collected for this study, however the patient pool of the hospital contains both Jewish- and Arab-Israelis.

When comparing the resistance rates of the studied Israeli isolates to other regions of the WHO, similar resistance rates for amoxicillin were observed in American regions and equivalent resistance rates for metronidazole were observed in European countries. However, the Israeli clarithromycin rates are higher than those of all regions and levofloxacin rates are lower than those of all regions [4].

The main limitation of this study is the relatively small sample size and possible limited generalizability both in the Middle East and globally, as isolates originate from a single hospital in Israel. Furthermore, the isolates were de-identified, and as such we were unable to characterize isolates on demographic factors like age, sex, ethnicity, or immigration status. Moreover, all isolates were recovered as part of routine diagnosis, likely from patients who failed several treatment lines, and also were not tested by reference dilution methods. No information about first line treatment antimicrobial resistance rates was collected, since primary diagnosis was made with non-invasive tests (e.g., urea breath test (UBT) or stool antigen test). Additionally, as *H. pylori* is a fastidious bacterium, it is possible that the cultured Israeli isolates are somehow different from those which failed to be cultured in a clinical laboratory setting. Thus, further studies should corroborate our findings.

## 5. Conclusions

In conclusion, in light of the growing interest in using WGS for routine microbiological diagnosis [41], and the future possibility of harnessing metagenomic NGS for analyzing infections caused by fastidious bacteria such as *H. pylori*, the challenges in inferring phenotypic AST from genomic analysis shown in our resistome analysis should be addressed in order to underpin the future integration of NGS in the study of this microorganism in the clinical setting.

## Figures and Tables

**Figure 1 microorganisms-09-00002-f001:**
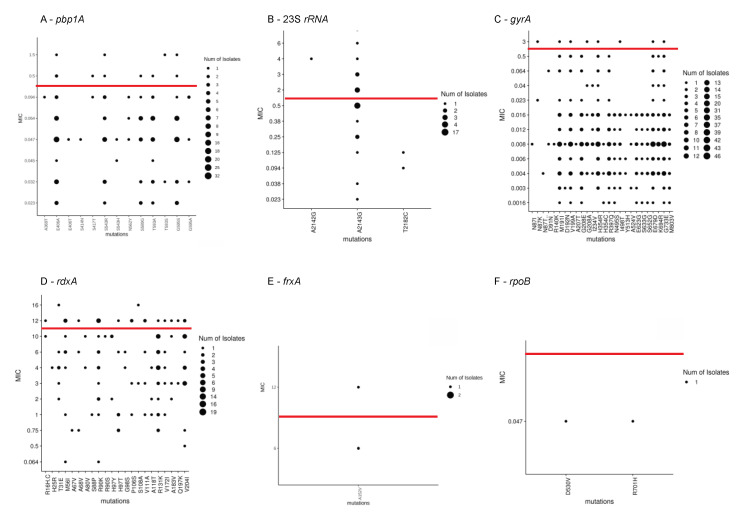
Concordance between minimum inhibitory concentration (MIC) values and genotypic mutations in (**A**) *pbp1A*, (**B**) 23S *rRNA*, (**C**) *gyrA*, (**D**) *rdxA*, (**E**) *frxA*, and (**F**) *rpoB* detected in Israeli isolates (*n* = 48). Red lines reflect the resistance breakpoint per British Society for Anti-Microbial Chemotherapy (BSAC) guidelines.

**Figure 2 microorganisms-09-00002-f002:**
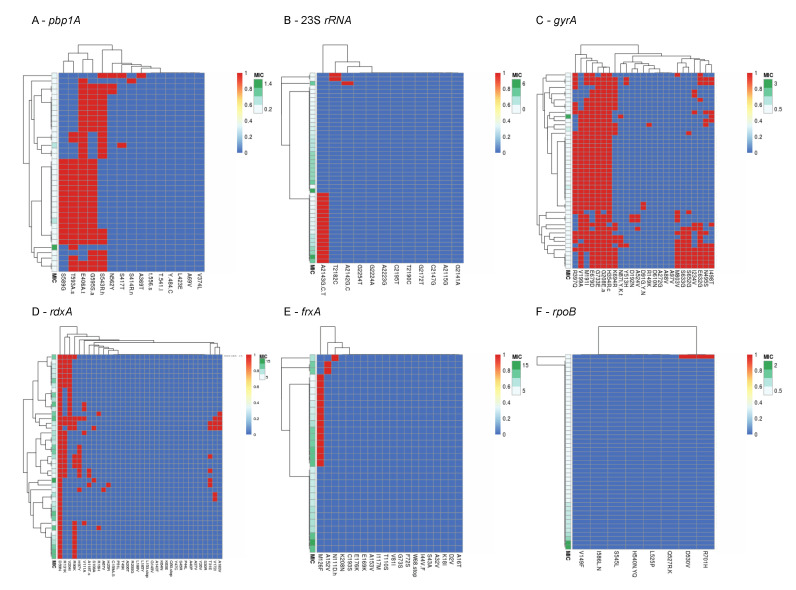
Visualization of clustering of mutations in (**A**) *pbp1A*, (**B**) 23S *rRNA*, (**C**) *gyrA*, (**D**) *rdxA*, (**E**) *frxA*, and (**F**) *rpoB* likely associated with phenotypic resistance among Israeli isolates, compared to the phenotype (MIC values). The clustering of mutations (top of each panel) reflects their co-existence in single genomes. Presence of mutations is denoted in red and absence in denoted in blue. MIC values are shown on a green scale.

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
