# Peer review of "Genomic Analysis of Antimicrobial Resistance Genotype-to-Phenotype Agreement in Helicobacter pylori"

_microorganisms, 2020, doi:10.3390/microorganisms9010002_

Round 1

Reviewer 1 Report

The aim of the work entitled “Genomic Analysis of Antimicrobial Resistance Genotype-to-Phenotype Agreement in Helicobacter pylori” was to determine the usefulness of WGS analysis in case of infections produced by H. pylori and checking their antibiotic resistance.

  • Please use the citation style compatible for MDPI journals
  • Please use the synonyms of the words: "differ/ differing/ different" (Introduction and Discussion) and "comparing / compared" (Discussion) -> to many of similar words in a close distance
  • Make clear in the text that "R" is resistant [Materials and Methods -> 2.3 Antimicrobial resistance-associated mutations analysis]
  • frxA gene -> without the underline [3.3. Novel point mutations]
  • The fragment of the text in lines 179-188 is to be improved because it is too detailed, and according to the reviewer, it should be more generalized
  • Please expand the explanation regarding the differences in the results obtained - the difference in the strain pool cannot explain everything (Discussion, lines 191-194)

Additionally what with results of others:

Microorganisms. 2020 Jun; 8(6): 887.

https://www.ncbi.nlm.nih.gov/pmc/articles/PMC7356661/

J Clin Med. 2019 Jan; 8(1): 53.

https://www.ncbi.nlm.nih.gov/pmc/articles/PMC6351930/

Why were these teams able to demonstrate the usefulness of this method? Please refer to these references.

  • “… those which failed to be cultured in a clinical laboratory setting” (Discussion, lines 206-207)

The reason for this phenomenon may be the morphological transformation into a coccoid form and difficulties in studying this form of H. pylori

Pathogens. 2020 Mar 4;9(3):184. doi: 10.3390/pathogens9030184.

https://pubmed.ncbi.nlm.nih.gov/32143312/

Please refer to this reference.

Author Response

We thank Reviewer 1 for their helpful comments on our manuscript. 

  • We have completed the formatting and grammar edits suggested by Reviewer 1 and updated the citation style.
  • “Please expand the explanation regarding the differences in the results obtained - the difference in the strain pool cannot explain everything (Discussion, lines 191-194)” – At line 197 we discuss that these differences may be due to sample size limitations, both in our study and others, however we have expanded this explanation to include different factors.
  • “Additionally what with the results of others” - We have incorporated the suggested citations and included them in our discussion at Line 159
  • “The reason for this phenomenon may be the morphological transformation into a coccoid form and difficulties in studying this form of H. pyloriWe have incorporated this suggested citation and integrated this concept into our discussion at line 174.

Reviewer 2 Report

This study reports the use of whole genome sequencing to comprehensively analyze the H. pylori resistome and phylogeny in order to characterise Israeli H. pylori.

I find this study interesting and deserving publication although I have some comments for the consideration of the authors.

Line 42 add the importance of know the percentage of resistance as reported by Leonardo H Eusebi et al. Epidemiology of Helicobacter pylori infection. Helicobacter . 2014 Sep;19 Suppl 1:1-5. doi: 10.1111/hel.12165.

Add the reference
The co-infection with other microrganisms and the therapy failure can potentiate the injury of gastric diseases as reported by Fasciana et al PhOL 2017. Underline the importance of vaccination above all in patients with co-infection, add the reference HELICOBACTER PYLORI AND EPSTEIN–BARR CO-INFECTION IN GASTRIC DISEASE Fasciana T., Capra G., Calà C., Zambuto S, Mascarella C., Colomba C., Di Carlo P., Giammanco A. PhOL 2017.

know the authors the eradication rates? If possible report the eradication rates.
There is difference of rate resistance among sex and age of patients?
Report this data if is possible.

Author Response

We thank Reviewer 2 for their helpful comments regarding our manuscript. 

  • “Line 42 add the importance of know the percentage of resistance as reported by Leonardo H Eusebi et al. Epidemiology of Helicobacter pylori infection. Helicobacter . 2014 Sep;19 Suppl 1:1-5. doi: 10.1111/hel.12165.” – We have integrated this citation into our discussion at line 201.
  • “The co-infection with other microrganisms and the therapy failure can potentiate the injury of gastric diseases as reported by Fasciana et al PhOL 2017. Underline the importance of vaccination above all in patients with co-infection, add the reference HELICOBACTER PYLORI AND EPSTEIN–BARR CO-INFECTION IN GASTRIC DISEASE Fasciana T., Capra G., Calà C., Zambuto S, Mascarella C., Colomba C., Di Carlo P., Giammanco A. PhOL 2017.” – We have integrated this citation into our introduction at line 50.
  • “Know the authors the eradication rates? If possible report the eradication rates. There is difference of rate resistance among sex and age of patients?
    Report this data if is possible.” These data were de-identified and thus we do not have access to these data. We have added further details regarding this to the limitations.

Reviewer 3 Report

Well described study with appropriate English and good documentation of the methods as well reporting. I suggest an acceptance along with my congratulations 

Author Response

We thank Reviewer 3 for their kind remarks and for taking the time to review our manuscript.

Round 2

Reviewer 1 Report

The authors have followed recommendation of reviewers. In my opition, it can be published now.